# Long-Term Efficacy and Toxicity of Intensity-Modulated Radiotherapy in Bulky Cervical Cancer

**DOI:** 10.3390/ijerph20021161

**Published:** 2023-01-09

**Authors:** Yu Wang, Tan-Tzu Lo, Lily Wang, Shih-Tien Hsu, Sheau-Feng Hwang, Chien-Hsing Lu, Lou Sun

**Affiliations:** 1Department of Gynecology and Obstetrics, Taichung Veterans General Hospital, Taichung 40705, Taiwan; 2Department of Obstetrics and Gynecology, Madou Sin-Lau Hospital, Tainan 72152, Taiwan; 3Department of Radiation Oncology, Taichung Veterans General Hospital, Taichung 40705, Taiwan; 4Center for General Education, Ling Tung University, Taichung 408284, Taiwan; 5School of Medicine, China Medical University, Taichung 404333, Taiwan; 6Department of Palliative Care Unit, Taichung Veterans General Hospital, Taichung 40705, Taiwan; 7Institute of Biomedical Sciences, Ph.D.Program in Translational Medicine, and Rong-Hsing Research Center for Translational Medicine, National Chung-Hsing University, Taichung 40227, Taiwan

**Keywords:** cervical cancer, intensity-modulated radiotherapy, concurrent chemoradiotherapy, cystitis, proctitis

## Abstract

Treatment of bulky cervical cancer is associated with both high adverse effects and local recurrence rates with traditional box method radiotherapy. Intensity-modulated radiotherapy (IMRT) has been adopted for the treatment of cervical cancer in order to deliver more precise radiation doses to the target region. We retrospectively enrolled a total of 98 patients with cervical cancer ≥4 cm who completed IMRT and point A-based brachytherapy treatment. The median follow-up time of the cohort was 6.84 years, with the 5-year OS and DFS being 66.33% and 75.12%, respectively. In addition, 7.14% of patients experienced local recurrence, 12.24% had distant recurrence, 6.12% had both local and distant recurrence, and 3.06% had persistent disease. In the univariate analysis, lymph node metastasis, higher creatinine levels, higher initial CA-125 and receiving chemotherapy other than cisplatin were all associated with a worse PFS. A tumor size ≥6 cm was associated with an increased incidence of higher grade of acute diarrhea. Grade 3 late radiation proctitis and cystitis developed in 11.22% and 13.27% of patients, respectively. The local recurrence rates and overall efficiencies were not inferior to other studies involving traditional pelvic external beam radiation therapy with concurrent chemotherapy. The safety and efficacy of IMRT for bulky cervical cancer were acceptable.

## 1. Introduction

Cervical cancer was the fourth most common cancer among women globally in 2020 [1]. Cisplatin adopted in concurrent chemoradiotherapy (CCRT) has been in use for over 20 years. Since the introduction of CCRT, survival rates in patients with locally advanced cervical cancer have significantly improved [2,3,4]. However, bulky cervical cancer, which is usually defined as a diameter of ≥4 cm, has shown poorer prognosis and local control rates [5]. Adjuvant hysterectomy following radiation was accepted in the 1960s, but soon after more evidence proved that this treatment offered no survival benefits and has since been abandoned [6,7]. Although improvements in survival rates for patients diagnosed with bulky tumors have been seen in recent years, high treatment failure and poor local control still remain a concern.

Whole Pelvic Radiation Therapy (WPRT) is commonly used in gynecologic malignancies, particularly radiosensitive cancers such as cervical cancer. To improve the conformality of dose distribution, intensity-modulated radiation therapy (IMRT), a more advanced 3D-conformal radiation therapy (CRT) planning system, has been developed. IMRT allows for a varying intensity of irradiation across the path of the treatment beam and delivers a more precise radiation dose to the target, while minimizing the dose to normal tissues. Compared with traditional CRT, IMRT is believed to lower the risk of side effects, including gastrointestinal (GI) toxicity and genitourinary (GU) toxicity [8]. However, clinical trials comparing IMRT and CRT have shown no definite increase in survival rates in many malignancies [9].

The American Society for Radiation Oncology (ASTRO) 2020 guidelines [7] for cervical cancer stated that for women receiving definitive RT with or without chemotherapy, IMRT is conditionally recommended to help decrease acute and chronic toxicity. As for women undergoing postoperative RT with or without chemotherapy, IMRT is strongly recommended to decrease acute and chronic toxicity. The National Comprehensive Cancer Network (NCCN) guidelines for cervical cancer, 2022 [10], offered the same recommendation. IMRT can be useful when high doses are required to treat gross disease. 

IMRT has been adopted as part of gynecologic cancer treatment since 2002 [11]. Studies have shown that IMRT can lower GI and GU adverse effects in post hysterectomy CCRT and locally advanced cervical cancer [12,13]. However, there is currently no literature focusing on IMRT outcomes and its side effects in bulky cervical cancer. In this paper, we analyzed the effectiveness and side effects of IMRT with CCRT treatment in bulky cervical cancer patients.

## 2. Materials and Methods

### 2.1. Patients

Our hospital began implementing IMRT for all cervical cancer patients who required radiotherapy in January 2004. A total of 1943 patients were diagnosed with cervical cancer between 2004 and 2014. One hundred four patients fulfilled the bulky cervical cancer requirement (tumor size ≥4 cm) and received CCRT treatment in our hospital. Among them, 4 did not complete radiotherapy in our hospital, while 2 patients were diagnosed with histological types of neuroendocrine tumor. A total of 98 bulky cervical cancer patients were then enrolled from 2004 to 2014 for this retrospective study. The inclusion criteria for participation was: (1) a pathological diagnosis of cervical cancer made in Taichung Veterans General Hospital (VGHTC), Taiwan, or another institute that had been clearly recorded on the patient’s medical chart; (2) diagnosis of a tumor size larger than 4 cm by a pelvic computed tomography (CT) scan, bimanual pelvic and rectal examination, sonography and/or magnetic resonance imaging (MRI); (3) an International Federation of Obstetrics and Gynecology (FIGO) 2014 staging criteria IB2 to IVA; (4) histological type, including squamous cell carcinoma, adenocarcinoma, carcinoma and clear cell carcinoma; (5) disease diagnosis from 2007 to 2014; radiotherapy using the IMRT technique; and (6) completion of IMRT. Exclusion criteria included: (1) patients who were pregnant; (2) a history of other malignancies within the past 5 years; (3) patients who did not complete radiotherapy in our hospital; and (4) those who underwent IMRT after surgery.

Amongst the 98 patients, 93 received brachytherapy, while 90 received concurrent chemotherapy. Medical history, image studies, pretreatment staging, blood cell tests, bimanual pelvic examinations and biopsy pathological reports were all reviewed. Every patient underwent a CT scan for pelvic and metastasis evaluation. For those whose metastasis was suspected or clinically needed, a positron emission tomography (PET) scan was arranged. Additionally, a whole-body bone scan was performed in patients with a risk of bone metastasis. Parametrial involvement was evaluated through a bimanual pelvic examination according to FIGO 2014 staging. Clinical lymph node metastasis criteria, including: (1) a CT scan showing enlarged lymph nodes with a short axis over 0.8 cm or central necrosis; (2) a PET scan showing an increased 2-deoxy-2-fluoro-D-glucose (FDG) uptake over the 2.5 standardized uptake value (SUV) compatible with an equivocal lymph node region CT scan image; and (3) an MRI showing enlarged lymph nodes over 0.8 cm and a heterogeneous enhanced signal intensity on a T1-weighted image. A biopsy was performed during a colonoscopy or cystoscopy if bladder or rectal invasion was suspected. The FIGO 2014 staging system was adopted for clinical staging. 

### 2.2. Radiotherapy

The planning system for IMRT involved the Eclipse (Varian Medical Systems, Palo Alto, CA, USA) software program. A dynamic multileaf linear accelerator with a photon energy of 10 MV was used for the IMRT treatment plan. A more detailed description of this technology is provided in our 2015 study [14]. 

For CT stimulation, a vacuum-fixed pad was used to immobilize the patients. A contrast-enhanced CT scan was performed using a radiopaque marker to define the cervix and vagina prior to contouring. Before IMRT treatment, each patient’s position and V-films were checked.

Gross tumor volume (GTV) was defined as the cervix tumor and uterus. Pelvic lymph nodes over 0.8 cm were defined as GTVn. IMRT clinical target volume (CTV) included a 0.5 to 1 cm margin to the GTV, the upper vagina, parametrium and lymph nodes. The planned target volume (PTV) with a 0.7–1 cm margin radially, superiorly and inferiorly was given to the CTV. Lymph nodes included the common iliac, internal iliac and external iliac lymph nodes (LN). Para-aorta (PA) lymph nodes were treated on an elective basis if enlarged PALNs were detected.

The treatment field was extended from the L4-5 interspace to 3 cm below the most distal cervical or vaginal site of disease. For patients with enlarged para-aortic lymph nodes, the treatment field’s superior field would extend to the T12-L1 interspace. The CTV of the para-aortic lymph nodes included the enlarged lymph nodes, aorta, inferior vena cava and 0.5–1 cm margin radially. The IMRT plan was as follows: (1) GTVn: 54–60 Gy in 30 fractions, (2) GTVp: 50.4–54 Gy in 28–30 fractions, and (3) CTV: 45–48 Gy in 25–30 fractions. The constraint for PTV was D_98_ > 98% of the prescribed dose. 

The dose constraint for normal tissues included the bladder, small bowel, rectum and colon. The dose constraints were (1) small bowel: less than 0% to 50 Gy (V50 Gy), less than 5% to 45 Gy (V45 Gy), less than 25% to 30 Gy (V30 Gy), less than 75% to 15 Gy (V15 Gy); (2) colon: less than 0% to 50 Gy (V50 Gy), less than 10% to 45 Gy (V45 Gy), less than 25% to 30 Gy (V30 Gy), less than 65% to 15 Gy (V15 Gy); (3) rectum: less than 0% to 55 Gy (V55 Gy), less than 30% to 50 Gy (V50 Gy), less than 60% to 40 Gy (V40 Gy), less than 75% to 30 Gy (V30 Gy); (4) bladder: less than 0% to 55 Gy (V55 Gy), less than 35% to 50 Gy (V50 Gy), less than 65% to 40 Gy (V40 Gy), less than 75% to 30 Gy (V30 Gy).

Iridium-192 was used in high dose rate (HDR) brachytherapy. A total of 20–30 Gy was given in two fractions a week, for a total of three to six times to point A. The median number of HDR brachytherapy was six fractions.

We suggested image-guided radiation therapy (IGRT) with daily cone beam CT (CBCT) for every patient. Repeat CT simulation and reduced portals were arranged after 20 fr. during radiotherapy for patients with bulky cervical tumor to adjust for the tumor regression and organ motion.

### 2.3. Chemotherapy

Cisplatin 40 mg/m^2^/week was routinely prescribed with dose modifications of 30 mg/m^2^/week if patients fulfilled the following criteria: (1) patients received extended field irradiation; (2) age over 70 years; (3) grade 1 or 2 hematological toxicity occurred during the previous cycle and (4) a Karnofsky performance status score ≤70. If poor renal function was noted pretreatment, cisplatin 20 mg/m^2^ plus fluorouracil 400 mg/m^2^ or daily oral tegafur 100 mg and uracil 224 mg (Ufur) three times a day would be prescribed.

Chemotherapy would be suspended if grade 3 or 4 hematological toxicity occurred. Dose modifications were prescribed weekly based on the acute toxicity grade.

### 2.4. Adverse Events

Acute adverse events during CCRT were assessed weekly using the Common Terminology Criteria for Adverse Events (CTCAE) v3.0. Long-term adverse events were assessed during follow-up visits.

### 2.5. Radical Surgery in Patients with Residual Disease after CCRT

We offered patients the option of surgery under two conditions. First, if patients showed a large residual tumor after IMRT, at the end of external beam irradiation courses, we offered patients the option to proceed with brachytherapy or hysterectomy. Second, if either a Pap smear, bimanual pelvic examination or sonography disclosed suspected residual disease 8 to 10 weeks after completing CCRT, a cervical biopsy would be performed. If the biopsy confirmed residual tumor cells and no evidence of metastasis, patients would undergo a surgery evaluation. The surgery would involve a radical or modified radical hysterectomy, bilateral salpingo-oophorectomy and lymphadenectomy.

### 2.6. Follow-Up Strategy

Persistent disease was defined as a tumor observed in 6 months, while recurrent disease was defined as a tumor detected after 6 months. Routine blood tests were conducted weekly during CCRT. After therapy had been completed, during the first three years of follow-up, the patient underwent a Pap smear, bimanual pelvic examination and blood test every 3 months, along with an abdominal CT every 6 months. From the third year to the fifth year, patients received a Pap smear, bimanual pelvic examination and follow-up blood tests every 6 months. After five years, patients received an annual follow-up Pap smear and bimanual pelvic examination. 

### 2.7. Statistical Analysis

The endpoints were disease-free survival (DFS) and overall survival (OS). OS was defined as the duration from the initial date of CCRT to the date of death or last follow-up. DFS was defined as the time from the initial date of CCRT to the date of evidence of local recurrence or distant metastasis. The Kaplan-Meier method was applied for survival analysis. A Cox proportional hazard model was used for age, lymph node metastasis, histology, pretreatment hemoglobin levels, cumulative chemotherapy dose, stage, CA-125, creatine and side effects for both univariate and multivariate comparison to estimate the 95% confidence intervals and hazard ratio. Statistical analyses were performed with SPSS software, version 20. A *p*-value < 0.05 was defined as statistically significant.

## 3. Results

A total of 98 patients fulfilled the inclusion criteria with a mean age of 56.7 years. Sixty-seven patients had a tumor size between 40 and 60 mm, while 31 had a tumor size ≥60 mm. The most common histologic type was SCC (85 patients). Seventy-nine patients received weekly cisplatin, with an average cisplatin cumulative dose of 195.34 mg/m^2^. Ninety-five percent of the patients received brachytherapy. Table 1 shows the patients’ demographic features and characteristics. Five patients who didn’t receive brachytherapy all underwent surgery after external beam irradiation. Table 2 illustrates the patients’ disease status and outcomes.

The median follow-up period was 6.84 years. The 2-year, 5-year and 10-year progression-free survival rates were 76.07%, 73.81% and 66.33%, respectively. The 2-year, 5-year and 10-year survival rates were 90.64%, 75.12% and 71.46%, respectively (Figure 1). No treatment-related death occurred during this study. A total of six patients received a post-CCRT radical hysterectomy, with four of them determined to be disease-free after surgery. In five of six patients who received surgery, large residual tumors were detected after external beam irradiation. At the end of external beam irradiation courses, all five patients chose to receive surgery. One patient was diagnosed with persistent disease without undergoing a radical hysterectomy. A total of 28 patients (28.57%) experienced tumor recurrence. The recurrence rate was higher in the adenocarcinoma patients (50%) when compared with squamous cell carcinoma patients (26.5%). The recurrence types are outlined in Table 2.

Table 3 summarizes the univariate and multivariate analyses of recurrence. Patients with lymph node metastasis, initial CA-125 levels over 34.86 U/mL and creatinine levels ≥1.3 mg/dL after treatment were associated with a higher recurrence rate. Cumulative cisplatin >180 mg/m^2^ was associated with a lower recurrence rate when compared with other chemotherapy agents, including cisplatin plus 5-FU and oral Ufur.

Table 4 summarizes the univariate analysis of death. Cumulative cisplatin > 180 mg/m^2^ was also associated with a better outcome when compared with other regiments.

Toxicities of IMRT are shown in Table 5. Thirty-four patients developed radiation proctitis, with 11 of them being grade 3. Twenty-four patients underwent a colonoscopy or sigmoidoscopy to confirm radiation proctitis. Twenty-eight patients suffered from radiation cystitis, including 13 with grade 3. Six patients developed rectovaginal fistula, while six received a colostomy, Hartmann procedure or segmental resection of the sigmoid colon. Three patients were diagnosed with ischemic bowel or bowel perforation. One patient received hyperbaric oxygen therapy and one patient was diagnosed with rectal cancer and expired eight years after her first radiation therapy session.

The analysis of the relationship between tumor size and side effects in Table 5 revealed a significant increase in events of anemia (≥grade 2) and diarrhea (≥grade 2) in patients with a tumor size ≥6 cm. 

Regarding concurrent chemotherapy, seventy-nine patients received concurrent weekly cisplatin 30–40 mg/m^2^ with a maximum of six doses. Among these patients, 47 received six cycles, 15 received five cycles, 10 received four cycles, 2 received three cycles, 3 received two cycles and 3 received one cycle. Five patients received weekly cisplatin 20 mg/m^2^ plus fluorouracil 400 mg/m^2^, while six received daily oral tegafur 100 mg and uracil 224 mg (Ufur) three times a day due to their old age and poor renal function. For 20 of the patients, it was necessary to lower their weekly cisplatin dose in order to complete CCRT. A total of 51 patients needed an adjustment to their initial cisplatin dose or had at least one dose of concurrent chemotherapy suspended due to side effects.

## 4. Discussion

In this study, the follow-up duration was longer than in previous studies. The focus of this study was bulky cervical cancer patients receiving IMRT treatment. Most patients also received concurrent chemotherapy along with high-dose rate brachytherapy. We demonstrated herein that bulky cervical cancer under IMRT with CCRT had a high hematologic toxicity, with anemia, leukopenia and thrombocytopenia occurring in 75%, 81% and 57% of patients, respectively. High incidence rates were also observed in all grades of radiation proctitis and radiation cystitis. However, severe proctitis and cystitis remained lower than 10% in all patients. Patients with bulky cervical cancer under CCRT with IMRT were able to achieve optimal OS and DFS rates. 

In early studies of CCRT with conventional radiotherapy, the 3-year OS ranged from 67% to 75%, with 83% being reported in one study which only included stage IB2 [2,3,15,16]. Previous studies have also shown that clinical outcomes, including DFS and OS, revealed no statistically significant differences between IMRT and conventional radiation [17,18]. The OS and DFS rates at 3 years were 80% and 74%, respectively, in our study. Although a previous study showed a poorer prognosis for bulky cervical cancer under the 4-field technique [19], our data revealed that under CCRT with IMRT, the DFS rates and OS were adequate.

Previous randomized controlled studies [2,3,4,15,20] have shown that under CCRT with conventional techniques, the local regional recurrence rate was 19 to 20%. Also, patients with locally advanced bulky cervical cancer experienced a high local recurrence rate due to inadequate irradiation to the bulky tumor [2,5,19]. Our study showed that recurrence types included 7 patients (7.14%) with local recurrence, 12 (12.24%) with distant metastasis, 6 (6.12%) with local and distant recurrence and 3 (3.06%) with persistent disease after CCRT and surgery treatment. Although the distant metastasis rate was higher, the local recurrence rate remained low. Under IMRT, the local recurrence rate remained low even in tumor sizes over 6 cm.

Six of our patients received a radical hysterectomy after CCRT due to persistent disease. One patient experienced persistent disease after surgery, and one patient had local recurrence. In previous studies, these patients had an extremely poor prognosis due to both radioresistance and chemoresistance. The use of salvage hysterectomy has been adopted since the 1990s [21,22,23,24,25]. A recent retrospective cohort study showed that when compared with systemic chemotherapy, salvage hysterectomy for persistent cervical cancer after receiving CCRT can reduce mortality rates by 60%. In our study, 67% of these patients achieved DFS after surgery.

Torres et al. [26] showed that a cisplatin dose >200 mg/m^2^ is an independent predictor of DFS. Our data showed a similar result in that patients under a cisplatin cumulative dose over 180 mg/m^2^ had a lower hazard ratio for recurrence when compared with a cumulative dose lower than 180 mm/m^2^ and other agents, as well as a lower hazard ratio for death when compared with other agents. Patients who received other chemotherapy agents, including 5FU and oral Ufur, had the worst outcomes. These results may be due to the fact that patients who opted to receive oral Ufur were mostly those diagnosed with multiple underlying diseases or were elderly. A cisplatin cumulative dose over 180 mg/m^2^ remains an essential part of CCRT treatment.

Our data showed higher CA-125 was associated with poor DFS. Elevated CA-125 was detected in 27% to 83% of cervical adenocarcinomas [27,28,29]. A study conducted in 2003 demonstrated CA-125 over 30 U/mL was an independent prognostic marker for OS in patients with cervical adenocarcinoma [30]. Our results revealed the same trend. The average values of CA-125 were 95.87 U/mL and 26.74 U/mL, respectively, in patients with adenocarcinoma and squamous cell carcinoma. The recurrence rates of adenocarcinoma and squamous cell carcinoma were 50% and 27%, respectively. Although adenocarcinoma comprised only 12% of patients in our study, these patients seemed to have a poorer outcome without significant differences. 

WPRT can cause a variety of adverse toxic effects. These toxicities may increase the risk of treatment break, lower quality of life and may even become life threatening. Compared with conventional techniques, which use blocks based on bone landmarks, IMRT was shown to be associated with lower GU and GI toxicity in previous studies due to its lower radiation dose at both the bladder and rectum [8,11,31,32]. For acute GI and GU toxicity, an early study showed significant decreases [8]. Mundt et al. demonstrated that patients in an IMRT group experienced lower GI toxicity (11.1% vs. 50%). However, 80% of the patients in this study were stage I or II cervical cancer. Another study which focused on cervical cancer stage IIB to IIIA also revealed lower acute GI toxicity (54.4% vs. 86.7%, *p* = 0.001) and lower acute GU toxicity (57.9% vs. 73.3%, *p* = 0.001) in IMRT the group [33]. In the latest randomized control study of cervical and endometrium cancer, approximately 33.7% of patients in the IMRT group complained of diarrhea, which was a lower percentage when compared with 51.9% in the conventional techniques group [34]. Regarding acute GI toxicity, our study revealed a high rate of diarrhea, with 65% of patients experiencing grade 1–2 diarrhea and 8% of patients reporting grade 3 diarrhea. However, in most cases, diarrhea could be controlled through medical treatment and ultimately subsided after completion of their CCRT courses. Grade 3 diarrhea was significantly higher in patients with tumor size over 6 cm, as we had only 3% of patients with tumor size 4–6 cm, compared with 20% of the patients with tumor size over 6 cm.

Regarding long-term GI and GU toxicity, Kidd et al. reported 6% grade 3 GU or GI toxicity [18] in an IMRT group, compared with 17% in a non-IMRT group. Our team revealed a proctitis rate of 1.4% and a cystitis rate of 9.2% in 2011 for all tumor sizes [12]. Hasselle et al. in 2011 reported a cystitis rate of 22% and a proctitis rate of 31% [35]. Our study included patients with more advanced staging and larger tumor sizes as compared with previous studies. Eleven percent and 13% of patients suffered from grade 3 proctitis and cystitis, respectively, while three patients received emergency surgery due to ischemic bowel or bowel perforation. These results indicate that in the patients with larger-sized tumors undergoing IMRT, although diarrhea was common during CCRT, long-term GU and GI toxicity were mostly grades 1 and 2, and only 10% would develop grade 3 toxicity. We speculate that toxicities in some patients might have occurred as a consequence of failing to follow dose constraints in organs at risk. It is challenging to deliver adequate doses to the CTV while keeping doses to the organs at risk as low as possible.

Hematologic toxicities are additional common adverse effects which may occur during CCRT. IMRT seems to offer no advantages regarding hematologic toxicity [36,37]. An early study showed that the hematologic toxicity rate was 57% in an IMRT group with early tumor stages [12]. Chen et al. [13] reported rates of a 69% for leukopenia, 28% for anemia and 9% for thrombocytopenia in locally advanced cervical cancer patients undergoing IMRT. However, another study reported that IMRT could lower hematologic toxicity risk [33]. A 2012 study reported the rate of leukopenia in the IMRT group was 17%, compared with 40% in the conventional techniques group. Although we lowered the initial cisplatin dose to 30 mg/m^2^/weekly if patients had fulfilled the necessary criteria, the rates of anemia, leukopenia and thrombopenia were still high, and therefore we often needed to discontinue or reduce the chemotherapy dose. High hematologic toxicity occurred in a total of 51 patients, resulting in an adjustment of the initial cisplatin dose or the suspension of at least one dose of concurrent chemotherapy during our study. In patients diagnosed with a larger tumor size, it seems that higher hematologic toxicity rates do occur more frequently and therefore there is a greater likelihood that the course of chemotherapy will be affected.

Studies suggested that novel techniques such as three-dimensional (3D) brachytherapy can lower GI and vagina toxicities [38,39,40,41]. 3D image-guided brachytherapy is an established standard of treatment in the definitive setting of cervical cancer and can achieve a higher dose to the high-risk CTV and potentially improve local control in patients with a bulky tumor [42]. 3D brachytherapy improves local relapse-free survival with potential impact on OS compared to point A-based brachytherapy [43]. An accurate image is more crucial in bulky cervical cancer. Therefore, patients with bulky cervical cancer might benefit from MRI-guided adaptive brachytherapy [44,45,46]. A prospective study from Japan focusing on the hybrid of intracavitary and interstitial brachytherapy (HBT) for locally advanced cervical cancer demonstrated that large volume was associated with increased risk of acute non-hematologic adverse events [47]. However, the majority of toxicities were less than grade 2. Therefore, HBT is a safe and promising novel technique. With higher rates of toxicities in bulky cervical cancer patients, 3D brachytherapy and HBT could be a possible solution. However, adoption of this approach is still limited in Taiwan as it is not covered by the National Health Insurance program.

Our study has several limitations. This was a retrospective study with a small number of patients, thus limiting the power of analysis. Second, due to a lack of patients receiving conventional radiotherapy, there were no comparisons made between efficacy and toxicity. Third, our treatment strategy was IMRT plus point A-based brachytherapy instead of 3D brachytherapy. Additionally, some analysis was limited due to varying doses of chemotherapy and radiotherapy between patients. A large, randomized multi-center study is still required in order to verify the effectiveness of IMRT in bulky cervical cancer patients.

## 5. Conclusions

Bulky cervical cancer under IMRT together with point-A based brachytherapy and CCRT can maintain low local recurrence rates. The toxicities were mostly grades 1 and 2 and could be resolved upon completion of treatment. The safety and efficacy of applying IMRT in bulky cervical cancer patients are acceptable.

## Figures and Tables

**Figure 1 ijerph-20-01161-f001:**
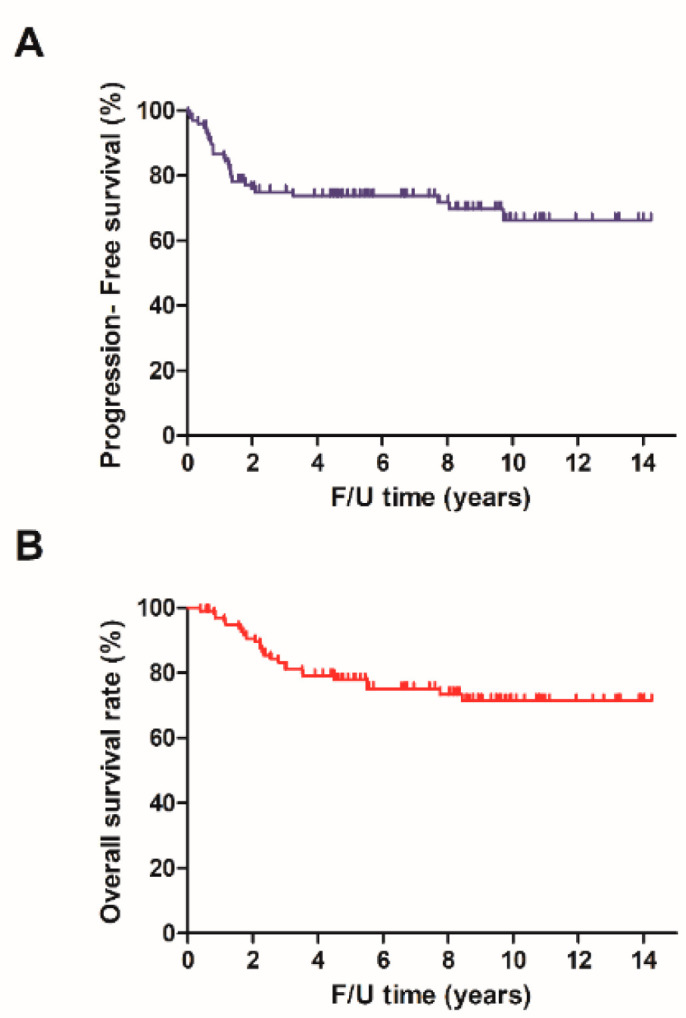
Disease-free survival (**A**) and overall survival (**B**).

**Table 1 ijerph-20-01161-t001:** Patient demographic features and characteristics.

		No. of Patients (Total of 98 Patients)
Age	<50	24 (24.49%)
	≥50	74 (75.51%)
Tumor size	40–<60 mm	67 (68.37%)
	≥60 mm	31 (31.63%)
Clinical stage	IB2	14 (14.29%)
	IIA2	1 (1.02%)
	IIB	38 (38.78%)
	IIIA	1 (1.02%)
	IIIB	43 (43.88%)
	IVA	1 (1.02%)
Underlying disease	Hypertension	25 (25.51%)
	Diabetes mellitus	14 (14.29%)
Pretreatment Hemoglobin	<11 g/dL	37 (37.76%)
	≥11 g/dL	61 (62.24%)
Creatinine during treatment period	<1.3 mg/dL	80 (81.63%)
	≥1.3 mg/dL	18 (18.37%)
Clinical lymph node metastasis	No	80 (81.63%)
	Yes	18 (18.37%)
Histology	Squamous cell carcinoma	83 (84.69%)
	Clear cell carcinoma	1 (1.02%)
	Adenocarcinoma	12 (12.24%)
	Carcinoma	2 (2.04%)
Differentiation	Well differentiated	6 (6.12%)
	Moderately differentiated	17 (17.35%)
	Poorly differentiated	68 (69.39%)
	Undifferentiated	1 (1.02%)
	Unrecorded	6 (6.12%)
Initial SCC-Ag ^a^	<4 ng/mL	39 (39.80%)
	≥4 ng/mL	59 (60.20%)
Initial CA-125	<40 U/mL	78 (79.60%)
	≥40 U/mL	20 (30.40%)

^a^ Squamous Cell Carcinoma Antigen.

**Table 2 ijerph-20-01161-t002:** Disease status and outcomes.

		No. of Patients (Total of 98 Patients)
Brachytherapy	No	5 (5.10%)
	Yes	93 (94.90%)
Concurrent chemotherapy	No	8 (8.16%)
	Cumulative cisplatin ≤180 mg/m^2^	36 (36.73%)
	Cumulative cisplatin >180 mg/m^2^	43 (43.88%)
	Cisplatin + 5FU	5 (5.10%)
	Oral Ufur	6 (6.12%)
Post CCRT ^b^ radical hysterectomy	Yes	6 (6.12%)
	No	92 (93.88%)
Outcome		
Recurrence	Total	28 (28.57%)
Recurrence type	Local recurrence	7 (7.14%)
	Local and distant recurrence	6 (6.12%)
	Distant recurrence	12 (12.24%)
	Persistent disease after CCRT and surgery	3 (3.06%)
	Persistence disease after CCRT but no recurrence after surgery (hysterectomy)	4 (4.08%)
Deaths		25
Follow-up years	(median)	6.84 (1.63–9.15) ^a^

^a^ 95% Cl. ^b^ Concurrent chemo-radiotherapy.

**Table 3 ijerph-20-01161-t003:** Univariate and multivariate analyses of recurrence.

	Simple Model			Multiple Model		
	HR	(95% CI)	*p*-Value	HR	(95% CI)	*p*-Value
Age (years)						
<50	1.00					
≥50	0.84	(0.37–1.91)	0.679			
Histology						
Squamous cell carcinoma	1.00					
Others	1.63	(0.66–4.03)	0.288			
Lymph node metastasis						
No	1.00			1.00		
Yes	2.30	(1.01–5.25)	0.047 *	3.22	(1.35–7.68)	0.008 **
Stage						
I + II	1.00					
III + IV	1.59	(0.75–3.34)	0.224			
Tumor size (mm)						
40–60 mm	1.00					
≥60 mm	1.54	(0.72–3.30)	0.262			
Creatinine during treatment						
<1.3 mg/dL	1.00			1.00		
≥1.3 mg/dL	3.02	(1.39–6.55)	0.005 **	3.00	(1.28–7.03)	0.012 *
Initial CA-125						
≤34.86 U/mL	1.00			1.00		
>34.86 U/mL	3.21	(1.50–6.86)	0.003 **	2.67	(1.19–6.02)	0.017 *
Chemotherapy						
^a^ Cisplatin > 180 mg/m^2^	1.00			1.00		
^a^ Cisplatin ≤ 180 mg/m^2^	2.36	(0.94–5.91)	0.068	2.35	(0.92–6.01)	0.075
Other	5.09	(1.59–16.34)	0.006 **	7.24	(1.37–15.03)	0.013 **
No	2.32	(0.60–8.98)	0.223	3.39	(0.45–7.96)	0.382
Pretreatment Hemoglobin						
<11	1.00					
≥11	0.77	(0.36–1.63)	0.491			
Differentiation						
Well and moderately differentiated	1.00					
Poorly differentiated	0.76	(0.34–1.69)	0.503			

Cox regression. * *p* < 0.05, ** *p* < 0.01, ^a^ cumulative dose.

**Table 4 ijerph-20-01161-t004:** Univariate analysis of death.

	Simple Model		
	HR	(95% CI)	*p*-Value
Age (years)			
<50	1.00		
≥50	2.77	(0.83–9.25)	0.098
Histology			
Others	1.00		
Squamous cell carcinoma	1.41	(0.53–3.76)	0.493
Grade			
Well and moderately differentiated	1.00		
Poorly differentiated	0.73	(0.31–1.69)	0.457
Lymph nodes			
No	1.00		
Yes	1.50	(0.60–3.75)	0.390
Stage			
I + II	1.00		
III + IV	1.59	(0.72–3.50)	0.250
Tumor size (mm)			
40–60 mm	1.00		
≥60 mm	1.01	(0.43–2.34)	0.985
Creatinine during treatment			
<1.3 mg/dL	1.00		
≥1.3 mg/dL	1.73	(0.72–4.15)	0.218
Initial CA-125			
≤34.86 U/mL	1.00		
>34.86 U/mL	1.19	(0.47–2.98)	0.712
Chemotherapy			
^a^ Cisplatin > 180 mg/m^2^	1.00		
^a^ Cisplatin ≤ 180 mg/m^2^	1.60	(0.60–4.30)	0.351
Other	5.66	(1.88–17.00)	0.002 **
No	2.79	(0.72–10.81)	0.137

Cox regression. ** *p* < 0.01, ^a^ cumulative dose.

**Table 5 ijerph-20-01161-t005:** Acute and late toxicity of intensity-modulated radiotherapy (IMRT) with concurrent chemotherapy (*n* = 98).

	Total (*n* = 98)		Tumor Size (mm)				*p*-Value
			40–<60 mm (*n* = 67)		≥60 mm (*n* = 31)		
	*n*	(%)	*n*	(%)	*n*	(%)	
Side effects							0.167
No	3	(3.06%)	1	(1.49%)	2	(6.45%)	
Yes	95	(96.93%)	66	(98.51%)	29	(93.55%)	
Nausea grade							0.152
Grade 1	17	(17.35%)	15	(22.39%)	2	(6.45%)	
Grade 2	24	(24.49%)	15	(22.39%)	9	(29.03%)	
Grade 3	1	(1.02%)	1	(1.49%)	0	(0.00%)	
Vomiting grade							0.803
Grade 1	10	(10.20%)	7	(10.45%)	3	(9.68%)	
Grade 2	12	(12.24%)	9	(13.43%)	3	(9.68%)	
Grade 3	1	(1.02%)	1	(1.49%)	0	(0.00%)	
Diarrhea grade (acute)							0.006 **
Grade 1	35	(35.71%)	26	(38.81%)	9	(29.03%)	
Grade 2	28	(28.57%)	23	(34.33%)	5	(16.13%)	
Grade 3	8	(8.16%)	2	(2.99%)	6	(19.35%)	
Hemoglobin grade							0.117
Grade 1	16	(16.33%)	12	(17.91%)	4	(12.90%)	
Grade 2	45	(45.92%)	32	(47.76%)	13	(41.94%)	
Grade 3	12	(12.24%)	5	(7.46%)	7	(22.58%)	
Leukopenia grade							0.139
Grade 1	17	(17.35%)	13	(19.40%)	4	(12.90%)	
Grade 2	22	(22.45%)	12	(17.91%)	10	(32.26%)	
Grade 3	40	(40.82%)	31	(46.27%)	9	(29.03%)	
Thrombocytopenia grade							0.142
Grade 1	48	(48.98%)	39	(58.21%)	9	(29.03%)	
Grade 2	5	(5.10%)	2	(2.99%)	3	(9.68%)	
Grade 3	2	(2.04%)	2	(2.99%)	0	(0.00%)	
Grade 4	1	(1.02%)	1	(1.49%)	0	(0.00%)	
Radiation proctitis grade (long term)							0.386
Grade 1	14	(14.29%)	10	(16.39%)	4	(12.90%)	
Grade 2	9	(9.18%)	5	(7.46%)	4	(12.90%)	
Grade 3	11	(11.22%)	8	(11.94%)	3	(9.68%)	
Radiation cystitis grade (long term)							0.624
Grade 1	8	(8.16%)	6	(8.96%)	2	(6.45%)	
Grade 2	7	(7.14%)	4	(5.97%)	3	(9.68%)	
Grade 3	13	(13.27%)	10	(14.93%)	3	(9.68%)	
Rectovaginal fistula ^a^							1.000
No	92	(93.88%)	63	(94.03%)	29	(93.55%)	
Yes	6	(6.12%)	4	(5.97%)	2	(6.45%)	

Chi-Square test. ^a^ Fisher’s Exact test. ** *p* < 0.01.

## Data Availability

Not applicable.

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
