# Peer review of "Long-Term Efficacy and Toxicity of Intensity-Modulated Radiotherapy in Bulky Cervical Cancer"

_ijerph, 2023, doi:10.3390/ijerph20021161_

Round 1

Reviewer 1 Report

The authors reported their retrospective experience with  Intensity-modulated Radiotherapy about the treatment of bulky advanced cervical cancer. I don't find any innovation that makes the manuscript interesting. In the formula in which it is presented seems to me anachronistic. I advise the authors to rewrite this retrospective toxicity outcomes about their experience. The manuscript requires a major revision.

Author Response

Thank you for your helpful suggestions. Our study mainly focuses on the long-term efficacy and toxicity of IMRT in bulky cervical cancer patients. We had applied 3D image-guided brachytherapy for several cases in our hospital under the protocol of one clinical trial, but the results are still unreleased. Under the current database and most patients' treatment choices in our hospital, it is difficult to rewrite the article with other data. We hope that we can produce another topic of cervical cancer in the near future.

Reviewer 2 Report

Long-term efficacy and toxicity of intensity-modulated radiotherapy in bulky cervical cancer submitted by Yu Wang et al,  is an extremely important topic of research and treating bulky cervical cancer with minimum normal toxicity and side effects are of prime concern. The treatment of bulky Cervical cancer employing IMRT is subject of intensive research and its has shown promise specifically with 3 D brachytherapy. However, in this retrospective study authors have attempted to compare IMRT Vs conventional radiation and its side effects. They also tried to compare IMRT outcomes and its side effects and  analyzed the effectiveness and side effects of IMRT with CCRT treatment in 64 bulky cervical cancer patients. There are several gaps and drawbacks in designing this study and few of them are cited below.

·        We  demonstrated herein that bulky cervical cancer under IMRT with CCRT had a high hematologic toxicity, with anemia, leukopenia and thrombocytopenia occurring in 75%, 81%, and 57% of patients, respectively. High incidence rates were also observed in all grades of radiation proctitis and radiation cystitis. However, severe proctitis and cystitis remained lower than 10% in all patients. Patients with bulky cervical cancer under CCRT with IMRT were able to achieve optimal OS and DFS rates. The above statement is quite confusing and not clear

·        Author needs to compare IMRT with 2D Vs 3D brachytherapy and this will enhance the impact of this work.

·        Authors also stated that Bulky cervical cancer under IMRT together with point-A based brachytherapy and CCRT can maintain low local recurrence rates. The toxicities were mostly grades 1 and 2 and could be resolved upon completion of treatment. The safety and efficacy of applying IMRT in bulky cervical cancer patients are acceptable. The above statement is not very clear and there are very significant toxicities involve in IMRT together with point-A based brachytherapy and CCRT and authors should describe in more details.

·        This study could be significantly improved if authors only focus on impact of IMRT with and without CCRT, IMRT Vs conventional RT and related toxicities and side effects and should also include dose information. Authors should also discuss IMRT with 3 D brachytherapy which is quite promising in treating bulky cervical cancer,

Author Response

Thank you for your helpful suggestions. 
We had applied 3D image-guided brachytherapy
for several cases in our hospital under the protocol of one clinical trial, but the results are still unreleased. There were few patients received conventional RT during our study period. Therefore, we might need more data or data from different hospital to perfrom such analysis. 

We try to analyzed the impact of CCRT but more than half of our patients needed an adjustment to their initial cisplatin dose or had at least 1 dose of concurrent chemotherapy suspended due to side effects.   
For dose analysis, we chose 12 patients from our study for analysis. Group 1 included 6 patients with tumor size 4-6 cm without proctitis nor cystitis. Group 2 included 6 patients all with tumor size over 6cm, 4 of them with grade 3 proctitis and/or cystitis, 2 of them with grade 3 proctitis and cystitis. The data didn’t show significant differences.  

Thanks for your time and we hope that in the near future we can rewrite the study with more data.